# Exploration of Gene Therapy for Alport Syndrome

**DOI:** 10.3390/biomedicines12061159

**Published:** 2024-05-23

**Authors:** Yafei Zhao, Qimin Zheng, Jingyuan Xie

**Affiliations:** 1Department of Nephrology, Ruijin Hospital, Shanghai Jiao Tong University School of Medicine, Shanghai 200025, China; zhaoyafeicn@126.com (Y.Z.); qimin_zheng@126.com (Q.Z.); 2Institute of Nephrology, Shanghai Jiao Tong University School of Medicine, Shanghai 200025, China

**Keywords:** Alport syndrome, gene therapy, recombinant adeno-associated virus

## Abstract

Alport syndrome is a hereditary disease caused by mutations in the genes encoding the alpha 3, alpha 4, and alpha 5 chains of type IV collagen. It is characterized by hematuria, proteinuria, progressive renal dysfunction, hearing loss, and ocular abnormalities. The main network of type IV collagen in the glomerular basement membrane is composed of α3α4α5 heterotrimer. Mutations in these genes can lead to the replacement of this network by an immature network composed of the α1α1α2 heterotrimer. Unfortunately, this immature network is unable to provide normal physical support, resulting in hematuria, proteinuria, and progressive renal dysfunction. Current treatment options for Alport syndrome include angiotensin-converting enzyme inhibitors and angiotensin receptor blockers, which aim to alleviate glomerular filtration pressure, reduce renal injury, and delay the progression of renal dysfunction. However, the effectiveness of these treatments is limited, highlighting the need for novel therapeutic strategies and medications to improve patient outcomes. Gene therapy, which involves the use of genetic material to prevent or treat diseases, holds promise for the treatment of Alport syndrome. This approach may involve the insertion or deletion of whole genes or gene fragments to restore or disrupt gene function or the editing of endogenous genes to correct genetic mutations and restore functional protein synthesis. Recombinant adeno-associated virus (rAAV) vectors have shown significant progress in kidney gene therapy, with several gene therapy drugs based on these vectors reaching clinical application. Despite the challenges posed by the structural characteristics of the kidney, the development of kidney gene therapy using rAAV vectors is making continuous progress. This article provides a review of the current achievements in gene therapy for Alport syndrome and discusses future research directions in this field.

## 1. Overview of Alport Syndrome 

Alport syndrome (AS) is a prevalent hereditary kidney disease characterized by a mutation in the gene responsible for encoding collagen IV (COL4), the primary component of the glomerular basement membrane (GBM). This mutation disrupts the normal synthesis and secretion of COL4, resulting in the formation of an abnormal GBM [1]. The GBM is an essential extracellular matrix component located between podocytes and endothelial cells, playing a key role in the function of the glomerular filtration barrier. COL4 constitutes approximately 50% of the total mass of the GBM and is vital for its stability and normal function [2,3]. Typically, COL4 in the GBM forms a heterotrimer composed of three peptide chains: α3, α4, and α5, the majority of which are produced by podocytes. They self-assemble into a triple helical conformer through interacting at their C-terminal NC1-domains, and are then secreted to contribute to the structure of the GBM. This also explains why the C-terminal NC1-domain is crucial for the normal function of COL4 [4]. However, mutations in the *COL4A3*, *COL4A4*, and *COL4A5* genes, responsible for editing the α3, α4, and α5 peptide chains, respectively, prevent the formation of the Col4α3α4α5 trimer. Instead, the immature GBM is partially replaced by the Col4α1α1α2 trimer, which lacks the same support strength as the Col4α3α4α5 trimer [5,6]. This structural alteration in the GBM leads to stratification, thickening, reel-like basket changes, and barrier dysfunction [1,7]. Consequently, individuals with mutations in the *COL4A3*, *COL4A4*, or *COL4A5* gene will continue to produce incorrect peptide chains, continually exacerbating the pathological changes in the glomerular basement membrane and podocyte injury.

The reported incidence of AS ranges from 1 in 5000 to 1 in 53,000 individuals [8,9,10,11,12]. Analysis of the genomic aggregation database reveals that the mutation rate of *COL4A5* is at least 1 in 2320, while that of *COL4A3/4* is at least 1 in 106 [12]. Based on the largest whole exome sequencing (WES) study in CKD to date, mutations in the *COL4A3/4/5* genes account for 30% of chronic kidney disease cases with single gene mutations, making it the second most common hereditary kidney disease after polycystic kidney disease, which represents 31% of cases [13]. Similarly, several studies have shown that end stage kidney disease (ESKD) patients with unknown reason often have underlying AS [13,14]. Furthermore, a significant number of COL4 mutations have been identified in patients with FSGS, indicating the importance of closely monitoring these individuals [15]. 

The pathogenesis of AS is intricate, and current treatment options are unsatisfactory. The precise pathogenic mechanism resulting from COL4 mutations had been poorly characterized until the vital research conducted by Pieri M’s group. Their work found that endoplasmic reticulum stress (ERS) and unfolded protein response constituted central pathogenic mechanisms in various conditions [16]. Furthermore, abnormal COL4 deposition, such as COL4α1α1α2, can initiate an inflammatory response and result in podocyte injury by activating discoidin domain receptor 1 (DDR1) and integrin α2β1 [17,18]. Simultaneously, cell injury can lead to disruptions in energy metabolism, dysregulation of NADPH oxidase, and the accumulation of lipids, ultimately causing podocyte lipotoxic injury and apoptosis that exacerbates kidney injury [19,20,21]. In the studies mentioned above, the inflammatory response was identified as a critical factor. Research has shown that activating Nrf2 can boost the body’s ability to combat oxidative stress and shield cells from damage, ultimately slowing down the advancement of renal fibrosis [22]. This highlights the potential role of bardoxolone methyl in this process. While the outcomes may not have met expectations, this effort remains a significant and valuable endeavor in the field of medical research [23]. A multitude of factors contributes to the progression of AS patients to ESKD, resulting in a poor prognosis. Treatment options for AS are currently limited, primarily relying on angiotensin-converting enzyme inhibitors (ACEIs) and angiotensin II receptor blockers (ARBs) as the standard treatments available. Although these medications can alleviate renal damage and postpone the progression to ESKD by multiple effects, such as decreasing intraglomerular pressure, their efficacy is limited, and it is imperative that treatment be initiated during the early stages of the disease in order to optimize its therapeutic benefits and prolong the lifespan of the kidney [24]. There is a growing body of experimental evidence supporting the exploration and development of new targets and mechanisms for AS treatment. Focused on ERS, we have found that excessive ERS and ERS-induced apoptosis play a significant role in podocyte injury, which could be mitigated through intervention in the proteasome pathway [25]. Other promising avenues include sodium-glucose cotransporter 2 inhibitors, endothelin receptor antagonists, aldosterone antagonists, and antioxidant inflammation modulators, which aim to mitigate AS tissue damage from various angles [26]. Additionally, the emergence of gene therapy drugs in clinical practice offers hope for improving the current landscape of AS treatment through genetic interventions.

## 2. What Are Gene Therapy and Gene Therapy Vectors?

Gene therapy holds the promise of curing diseases that are currently untreatable with conventional medications. It involves, but is not limited to, the use of genetic material to address or prevent diseases by manipulating specific genes within target cells through precise techniques. This can result in the alteration of gene function or the implementation of gene editing programs within cells to facilitate internal genetic modifications [27]. Current gene editing technologies primarily focus on somatic cells rather than germline cells, aiming to treat diseases within the patient’s body while avoiding the transmission of altered genes to future generations and the associated ethical dilemmas. The key aspect of gene therapy lies in delivering genetic material, such as DNA or RNA, to target cells. Once inside the cell, this genetic material can carry out its intended function, either by regulating gene expression or altering cellular characteristics to combat disease. In contrast to traditional treatments involving proteins or small molecule drugs, gene therapy has the potential to provide sustained protein expression for extended periods without the need for frequent infusions. Essentially, the correct genetic material can be introduced into the patient’s body to achieve a lasting therapeutic effect in a single treatment session [28,29].

In order to facilitate the transfer of genetic material into target cells, the use of vectors is essential. While plasmid DNA or RNA can be utilized for gene therapy, they lack the ability to target and enter cells to carry out normal functions without the assistance of vectors. Therefore, the use of specific vectors is essential for successful gene therapy delivery. Vectors, both viral and non-viral, are essential tolls in the transport process (Table 1). Viral vectors, in particular, have undergone significant advancements to meet the requirements of gene therapy applications. They offer high transfection efficiency and the capability to express target genes within specific cells, making them preferred options for clinical gene therapy [30]. Among the commonly used viral vectors are adenovirus, recombinant adeno-associated virus (rAAV), and lentivirus. Adenovirus, characterized by its high capacity to carry exogenous gene fragments and its broad spectrum of cell infection, was the first DNA virus developed for therapeutic purposes. It boasts genetic stability and significantly fewer side effects compared to traditional chemotherapy drugs, rendering it valuable in treating certain tumor diseases and preventing infections through vaccination. However, adenovirus does present certain limitations. Its high immunogenicity can trigger an immune response in the subject, potentially leading to fatal inflammation. Additionally, the duration of gene expression in the body is relatively short-lived. Overcoming these challenges is crucial for advancing the development of effective therapeutic drugs using adenoviral vectors [31,32,33,34,35]. Lentivirus, being a retrovirus with a particle size of around 100 nm, presents a significant challenge in clinical applications due to its tendency for homologous recombination and integration into the host chromosome. This integration can lead to unforeseen complications such as myelodysplastic syndrome or acute leukemia [36,37,38]. While advancements, such as the fourth-generation lentivirus, have addressed some issues related to homologous recombination, limitations remain regarding vector titer and the difficulty in effectively targeting infected cells, thereby limiting their clinical applicability [39,40,41]. Currently, lentiviral vectors are primarily utilized for in vitro gene therapy applications, particularly in tumor immunotherapy. This approach allows for targeted transfection and the screening of cells for any pathogenic homologous recombination events, thereby reducing the risk of severe diseases [42,43]. On the other hand, recombinant adeno-associated virus (rAAV), commonly known as AAV, was initially discovered as a contaminant in adenovirus preparations. Its genome consists of single-stranded DNA approximately 4.7 kb in length, with a particle size of around 25 nm [44]. AAV are replication-defective parvoviruses that typically require coinfection with a helper adenovirus or herpesvirus to efficiently infect cells, necessitating the use of a helper plasmid. AAV itself is non-pathogenic and has not been linked to any known human diseases. Moreover, it demonstrates low immunogenicity and exists in various natural serotypes, each with different tissue affinities, making it ideal for targeted transfection. Upon entering the host cell, the genetic material carried by AAV forms concatemers in the nucleus without integrating into the host genome, enabling long-term expression that could last from months to years [45]. Despite these favorable characteristics, several limitations hinder its clinical application, including its small capacity, potential neutralizing antibodies in some patients (particularly against AAV2 serotypes), low virus production efficiency, and high costs. Furthermore, animal studies have also reported severe toxicity following high-dose AAV9 administration [46,47,48,49,50]. Addressing these challenges is essential for advancing AAV as a viral vector for widespread use. Strategies to overcome issues such as capacity constraints, immune responses, production efficiency, and toxicity must be carefully considered in clinical research to ensure the safe and effective application of AAV-based therapies. Interestingly, recent reports have highlighted the use of messenger RNA (mRNA) gene therapy delivered through lipid nanoparticles (LNPs) as a promising method to assess the safety and effectiveness of gene therapy agents in clinical trials [51]. This approach offers a promising alternative to traditional viral vectors.

## 3. Available Attempts at Gene Therapy for Alport Syndrome

The key factor in the development of AS lies in mutations of one or more of the *COL4A3/A4/A5* genes. These mutations result in the abnormal synthesis and folding of the α3/α4/α5 peptide chain. This leads to the formation of abnormal COL4α3α4α5 trimers that are unable to support the COL4 skeleton network. The ultimate objective of gene therapy is to introduce healthy *COL4A3/A4/A5* genes into podocytes to replace the mutated genes, thereby producing a normal functional peptide chain to form COL4 and restore the function of the GBM. Reducing the synthesis of abnormal α3/α4/α5 peptide chains through DNA or RNA transduction could mitigate cellular inflammation and damage, presenting an alternative gene therapy strategy. While the realization of this concept for clinical application is currently hindered by technological and material obstacles, some research studies have explored the potential of gene therapy in tackling AS.

The groundbreaking discovery of gene therapy has spurred extensive research into transducing target genes into kidney cells for the treatment of diseases. The interaction among *COL4A3/A4/A5* gene expression is pivotal for implementing AS gene therapy. In 2000, a team from France confirmed the transcriptional activity of *COL4A3* and *COL4A4* genes in podocytes of AS patients with *COL4A5* mutation, regardless of mutation type. This study underscores the potential for gene therapy to target specific mutated genes associated with AS, rather than transducing all genes linked to COL4 simultaneously [52]. In a notable experiment, adenovirus containing full-length *COL4A5* cDNA was transfected into a *COL4A5* nonsense mutation-induced AS model dog. The transfected smooth muscle cells successfully expressed α5, aiding in extracellular secretion and COL4 localization to the basement membrane [53]. As AS is hereditary, the effectiveness of acquired treatments in reversing the disease phenotype is crucial. This was addressed in an experiment using an inducible *COL4A3* mouse model, where *COL4A3* expression was induced at various time points in *COL4A3*−/− mice using a doxycycline induction system. The results showed the correction of abnormal GBM in postnatally induced *COL4A3*−/− mice, leading to significantly decreased (or no) proteinuria compared to *COL4A3* knockout mice succumbing to renal failure and death by 8 weeks of age. Notably, normal GBM was observed in some mice even at 23 weeks of age. These findings establish a robust pathophysiological foundation and open up possibilities for the clinical application of gene therapy in AS treatment [6].

The research team led by P. Heikkila in Finland conducted pioneering experiments involving adenovirus-mediated gene transduction in isolated human glomeruli in vitro. They successfully expressed the reporter gene, demonstrating the potential for gene therapy in disease treatment. Expanding on this achievement, the team proceeded to perfuse the porcine renal artery for 2 h in vivo, achieving an impressive 85% reporter gene expression rate in glomerular cells [54]. Subsequently, they endeavored to transfer the *COL4A5* gene using adenovirus in pigs, successfully expressing the α5 chain in porcine glomeruli and depositing it in the GBM. However, limitations, such as short duration of gene expression, strong immunogenicity, adverse reactions, and the inability to repeat drug administration, were noted. Despite these limitations, the study provides valuable insights into the potential of gene therapy, instilling confidence in its feasibility. While not definitively proving the therapeutic benefits of gene therapy, the study lays a solid foundation for further research and development in this promising field [55].

In addition to directly supplementing the expression of the deficient COL4 peptide chain in vivo, researchers have explored various indirect methods of gene therapy. One such method involves targeting renal microRNA-21 (miRNA-21), a molecule known to promote inflammation and cellular fibrosis. Studies have shown that levels of miRNA-21 are significantly elevated in *COL4A3*−/− AS mice. To counteract the effects of miRNA-21, an anti-miRNA-21 oligonucleotide has been developed. This oligonucleotide has proven effective in reducing renal fibrosis, lowering proteinuria levels, preserving renal function, and increasing the survival rate of mice in experimental models. These findings suggest that oligonucleotide therapy targeting miRNA-21 may hold promise for treating AS and related conditions [56]. Recent domestic studies have confirmed the elevated expression of miRNA-21 in the kidney tissue of patients with AS. This increased expression of miRNA-21 is positively correlated with urine protein levels, serum creatinine levels, and the severity of renal histological injury, further establishing miRNA-21 as a potential therapeutic target for AS [57]. However, a phase II trial of lademirsen, an anti-miRNA-21 agent, was prematurely terminated after an interim analysis revealed no significant improvement in renal function among patients treated with lademirsen compared to those given a placebo (NCT02855268; interim analysis data were collected from 24 patients who completed the 24-week double-blind treatment period). This setback dealt a blow to the progress of new drug development for AS. On a more positive note, the team led by Nozu K proposed an innovative approach by studying Alport mice with a *COL4A5* nonsense mutation. By utilizing an antisense oligonucleotide (ASO) to target the splicing enhancer on the DNA exon containing the nonsense mutation site, they were able to induce skipping of the entire exon with the stop codon. This resulted in the transformation of a truncating mutation with a severe clinical phenotype into an in-frame deletion mutation. This modification led to reduced levels of urinary protein and creatinine, improved renal pathology, and partial restoration of α5 chain expression. This groundbreaking research offers a promising avenue for potential therapeutic interventions in AS and underscores the importance of continued exploration and innovation in the field of genetic disorders [58].

Gene editing is also a vital aspect of gene therapy, though its success rate in vivo is currently low, making practical application challenging. Nevertheless, pioneering researchers have developed a method to harvest podocytes from patients’ urine and administer them to individuals with AS, targeting podocytes bearing mutations in *COL4A3* and *COL4A5*. Employing an AAV vector to deliver the necessary genetic material, they were able to edit the mutant genes in vitro, achieving relatively efficient and stable gene correction rates of 44.2% and 58.8% for *COL4A3* and *COL4A5*, respectively [59,60]. In addition to direct gene editing, CRISPR/Cas9 has emerged as a therapeutic avenue utilizing a mutant variant, dead Cas9 (dCas9), with endonuclease defects. By activating the expression of the COL4A6 gene in podocytes, researchers prompted the production of COL4α5α5α6 trimers, thereby normalizing the structure and function of the GBM and alleviating disease symptoms. Similarly, inhibiting the expression of laminin subunit α2 in atherosclerosis using dCas9 bound to a transcriptional repressor showed promise in reducing podocyte injury and halting disease progression [61]. In conclusion, we have developed a pattern diagram integrating the comprehensive summary of the primary mechanisms of Alport syndrome and the gene therapies currently under investigation (Figure 1) and the main approaches of novel therapies with their results (Table 2).

## 4. Difficulties Facing Gene Therapy for Kidney Diseases

While research into AS gene therapy continues from various perspectives, progress in this field has been relatively constrained compared to other organs or tissues. One significant hurdle lies in the selection of vectors. Although adenovirus has been extensively studied, its strong immunogenicity poses challenges by trigging significant immune responses in subjects. Additionally, adenovirus-mediated gene transduction in vivo exhibits short-lived expression, necessitating regular intermittent administration. This presents a significant obstacle to its clinical application. Furthermore, the possibility of the random integration of the lentivirus genome in subjects raises concerns among clinical researchers. While rAAV demonstrates promise as a vector for clinical applications, its small gene load presents a drawback. The gene load of rAAV is limited to approximately 4.7 kb, which decreases to 2.7 kb after excluding the target gene expression regulatory sequence. This poses a challenge, as the protein-coding sequences of *COL4A3/A4/A5*, which are all approximately 5.0 kb, exceed the carrying capacity of rAAV [62]. In conclusion, while advancements in AS gene therapy are underway, challenges related to vector selection and gene load limitation must be addressed to improve the feasibility of clinical applications in this field. On the other hand, glomerular podocytes play a crucial role in expressing and secreting the COL4α3α4α5 trimer, making them the primary focus for AS gene therapy. However, targeting podocytes is challenging due to the restrictive glomerular filtration membrane. The glomerular basement membrane (GBM) typically permits only particles with a diameter of less than 10–50 nm and a molecular weight of less than 50 kD to pass through, with a charge barrier further hindering target gene transduction efficiency through circulation [63]. In a rat model, the use of an rAAV vector to introduce green fluorescent protein via renal artery injection resulted in expression only within the renal tubules, not the glomerulus. This suggests that renal artery injection may not be a practical method for the rAAV transduction of glomerular cells [64]. Therefore, to explore alternative approaches, an endothelium-specific inducible transgene system was employed to induce *COL4A3* expression in *COL4A3*−/− mice. Unfortunately, this did not lead to the detection of the COL4α3α4α5 trimer or the restoration of the Alport disease phenotype [65]. This outcome emphasizes the absence of specific factors regulating the expression of *COL4A3/A4/A5* genes in cells other than podocytes, reaffirming podocytes as the primary target for AS gene therapy.

Despite the challenges, there is still hope. While the intravenous administration of gene-therapy agents may be less effective in reaching glomerular podocytes, achieving efficient target gene transduction with higher viral doses is possible. Professor Saleem Ma’s team achieved successful gene transduction of podocin-targeting podocytes by administering a large virus dose into the tail vein of mice. This resulted in the successful expression of podocin in induced podocin knockout mice, leading to reduced levels of urinary protein, serum creatinine, and urea. Renal pathology also exhibited some degree of improvement [66]. However, their study also revealed that using higher doses of rAAV in mice triggered an immune response, and the tail vein injection of rAAV resulted in non-target organ and cell transduction. This indicates the necessity for further refinement in delivery methods, dosages, and viral targeting. Despite these challenges, this study signifies a significant advancement in gene therapy for monogenic hereditary kidney diseases targeting the kidney and podocytes. Furthermore, a new solution to the rAAV capacity limitation has emerged. The dual AAV system design concept involves dividing longer target genes at specific sites, packaging them into different rAAV vectors for simultaneous delivery, and recombining them in target cells through homologous recombination to form full-length target genes [67]. However, the dual AAV-mediated gene therapy system developed by Professor Yilai Shu and his team at Fudan University has shown promise in reversing bilateral deafness in Otof−/− mice, and it has demonstrated both efficacy and safety in clinical trials involving pediatric patients [68,69,70]. The successful application of this dual AAV system not only introduces a novel strategy for gene therapy but also instills confidence in addressing a broad spectrum of diseases in urgent need of genetic intervention. As well as the dual AAV system, the transduction of mRNA to target cells via nanoparticles has also been extensively researched. The advancement in LNP has led to a reduction in their cytotoxicity, with certain isoforms capable of being metabolized by the human body without accumulating. Zhou’s study has demonstrated that mice with α-Gal A-deficiency showed a long duration of substrate reductions in tissues and plasma after a single intravenous injection of the mRNA-encoding human α-Gal A carried via LNP [71]. Moreover, early clinical trials have further confirmed their safety as vectors [51]. However, there are still considerable drawbacks to using this approach for the kidney. Nanoparticles carrying genes may indeed face challenges passing through the GBM, such as the human α-Gal A mRNA carried via LNP, mentioned earlier, with a particle size of 80–100 nm. While techniques such as reverse perfusion, direct injection into the renal interstitial, and electroporation have demonstrated efficacy in targeting mRNA to the kidney and treating glomerulonephritis in animal models, electroporation risks damaging the kidney itself, with a low benefit–risk ratio. Similarly, retrograde perfusion and direct injection into the renal interstitial have been associated with severe pain and poor prognosis, potentially impacting patient compliance and limiting widespread applicability in clinical settings [72]. Hence, it is evident that there remains a substantial journey ahead on the research path.

## 5. Summary and Prospects

With the increasing awareness of health and the wide usage of genetic testing in clinics, there is a growing identification and diagnosis of individuals with AS. What was once a rare disease is now becoming more prevalent, highlighting the urgency to accelerate the development of targeted treatment approaches. Gene therapy emerges as a revolutionary treatment modality with the potential to cure this monogenic disorder. While gene therapy for kidney diseases is still in its nascent stages, the primary challenge lies in effectively delivering therapeutic gene fragments to renal target cells. Despite this hurdle, researchers are actively exploring various animal models, carriers, and delivery methods to advance the field. By prioritizing gene therapy agents that have demonstrated success in other organs or tissues, we can leverage existing knowledge and spur innovation in AS gene therapy. Through ongoing research and experimentation, our aim is to achieve a significant breakthrough in the treatment of kidney diseases, offering renewed hope to patients affected by these conditions.

## Figures and Tables

**Figure 1 biomedicines-12-01159-f001:**
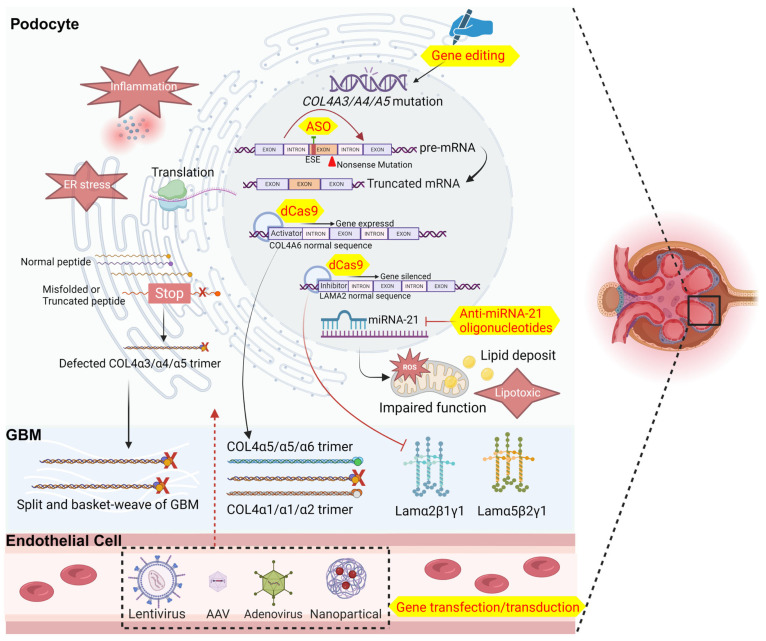
Pathogenesis of Alport and available gene therapies. dCas9, dead Cas9; ASO, antisense oligonucleotide; ESE, exon splicing enhancers; ROS, reactive oxygen species. The diagram was created with BioRender.com.

**Table 1 biomedicines-12-01159-t001:** Characteristics of gene therapy vectors.

Characteristic	AAV	Adenovirus	Lentivirus	Nanoparticle
Size	25 nm	60–90 nm	90–100 nm	Varies
Genome type	DNA (single-stranded)	DNA (double-stranded)	RNA (double-stranded)	DNA/RNA
Envelope	No	No	Yes	Yes
Genome size limitation	4.7 kb	~40 kb	~10 kb	Varies
Immunogenicity	Very low	Strong	Low	Low
Targeting	Yes	Yes	Yes	Yes
Integration	Rare (2%)	No	Yes	Moderate
Endurance	Months to years	Weeks	Months	Days to weeks

AAV, adeno-associated viruses; DNA, deoxyribonucleic acid; RNA, ribonucleic acid.

**Table 2 biomedicines-12-01159-t002:** Main approaches of novel therapies with their results.

Approach	Subjects	Treatment	Administration	Results	Reference
Gene transfection/transduction	AS dog	Adenovirus containing COL4A5	Bladder injection	α5 chain express in bladder smooth muscle and deposit in the basement membrane	[53]
	Normal pig	Adenovirus containing COL4A5	Renal artery perfusion	α5 chain express in porcine glomeruli and deposit in the GBM	[54,55]
Anti-miRNA-21 oligonucleotides	AS mice	anti-miRNA-21 oligonucleotide	Subcutaneous injection	Reduce renal fibrosis and urine protein, preserve renal function, and increase the survival rate of mice; but failed in clinic trial (NCT02855268)	[56]
ASO	AS mice	ASO	Subcutaneous injection	Reduce urinary protein and creatinine, improve renal pathology, and partial restore α5 chain expression	[58]
Gene editing by CRISPR/Cas9	Podocytes from AS patients	CRISPR/Cas9 carried by AAV	Cell transfection in vitro	Moderate gene correction rates	[59,60]
dCas9 targeting activator	HEK293T cells	CRISPRa/dCas9 carried by Lipofectamine 3000	Cell transfection in vitro	Replace the missing α3α4α5(IV) network with the α5α5α6(IV) network	[61]
dCas9 targeting inhibitor	HEK293T cells	CRISPRi/dCas9 carried by Lipofectamine 3000	Cell transfection in vitro	Reduce laminin α2 in the GBM	[26,61]

AS, Alport syndrome; COL4, collagen IV; GBM, glomerular basement membrane; RNA, ribonucleic acid; ASO, antisense oligonucleotide; CRISPR, clustered regularly interspaced short palindromic repeat; Cas9, CRISPR associated protein 9; AAV, adeno-associated virus; dCas9, dead Cas9.

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
