# Peer review of "Exploration of Gene Therapy for Alport Syndrome"

_biomedicines, 2024, doi:10.3390/biomedicines12061159_

Round 1
Reviewer 1 Report
Comments and Suggestions for Authors
The authors present a rather brief account of previous attempts to treat Alport syndrome with gene or gene related interventions. It is not the most comprehensive review but it touches upon the major publications of the past 20 years or so.
A major issue is that the References used in a big part of the Introduction seem to be randomly chosen, not reflecting the text content. This applies to refs 1-9 that I looked more carefully but also to others. For example, refs 1 & 2 refer to genotype-phenotype correlation and disease management respectively, not to the GMB content and structure.
Also, the first evidence of the unfolded protein response was published by Pieri M et al, JASN 2014. The cited refs 8-9 are important works but most inappropriate regarding the first works on the UPR in Alport syndrome.
In line 39, the authors state that the three alpha-collagen chains are polymerised. A better term is that the three chains self-assemble into a triple helical conformer through interacting at their C-terminal NC1-domains.
The text needs proofreading for the English language.
Comments on the Quality of English LanguageNeeds proofreading for the English language.
Author Response
Thank you very much for dedicating your time to reviewing this manuscript. We have uploaded our response, please see the attachment. Thank you again for your comments and suggestions.

Reviewer 2 Report
Comments and Suggestions for Authors
The review is a detailed description of gene therapy in Alport syndrome. There are some redundances and some experiments may be reported with less details. I would recommend to postpone chapter 1 dealing with vectors after chapter 2 and before chapter 4.
Author Response

(The authors gave the same response as above.)

Reviewer 3 Report
Comments and Suggestions for Authors
The review comperehensively demonstrates the present available and possible gene therapy-based interventions for the treatment of Alport syndrome, also discussing the major technical drawbacks and side-effects.
Grammar is fine, no further editing is required.
I just have minor suggestions to further improve the manuscript:
1) I was missing a graphical summary at the Conclusion section,
2) In my opinion a Table summarizing the main approaches with their results (even if negative) and associated references would be very helpful.
Author Response
Please provide a point-by-point response to the reviewer’s comments and either enter it in the box below or upload it as a Word/PDF file. Please write down "Please see the attachment." in the box if you only upload an attachment. An example can be found here.
